# Chemical activation of the mechanotransduction channel Piezo1

Ruhma Syeda[1,2], Jie Xu[3], Adrienne E Dubin[1], Bertrand Coste[1†], Jayanti Mathur[3], Truc Huynh[3], Jason Matzen[3], Jianmin Lao[3], David C Tully[3‡], Ingo H Engels[3§], H Michael Petrassi[3], Andrew M Schumacher[3], Mauricio Montal[2], Michael Bandell[3]*, Ardem Patapoutian[1]*

[1]Department of Molecular and Cellular Neuroscience, Howard Hughes Medical Institute, The Scripps Research Institute, La Jolla, United States; [2]University of California, San Diego, La Jolla, United States; [3]Genomics Institute of the Novartis Research Foundation, San Diego, United States

**Abstract** Piezo ion channels are activated by various types of mechanical stimuli and function as biological pressure sensors in both vertebrates and invertebrates. To date, mechanical stimuli are the only means to activate Piezo ion channels and whether other modes of activation exist is not known. In this study, we screened ~3.25 million compounds using a cell-based fluorescence assay and identified a synthetic small molecule we termed Yoda1 that acts as an agonist for both human and mouse Piezo1. Functional studies in cells revealed that Yoda1 affects the sensitivity and the inactivation kinetics of mechanically induced responses. Characterization of Yoda1 in artificial droplet lipid bilayers showed that Yoda1 activates purified Piezo1 channels in the absence of other cellular components. Our studies demonstrate that Piezo1 is amenable to chemical activation and raise the possibility that endogenous Piezo1 agonists might exist. Yoda1 will serve as a key tool compound to study Piezo1 regulation and function.

*For correspondence: mbandell@gnf.org (MB); ardem@scripps.edu (AP)

Present address: †Centre de Recherche en Neurobiologie et Neurophysiologie de Marseille, Centre national de la recherche scientifique, Aix Marseille Université, La Jolla, United States; ‡Novartis Institutes for Biomedical Research, Emeryville, United States; §Department of Pharmaceutical Sciences, Appalachian College of Pharmacy, Oakwood, United States

Competing interests: The authors declare that no competing interests exist.

## Introduction

Mechanotransduction describes processes by which mechanical forces are converted into biological responses. Mechanotransduction is essential for physiological functions including the sense of touch, hearing, and blood pressure regulation. The molecular mechanisms involved in mechanotransduction have been largely unknown, but mechanically activated cation channels are postulated to play important roles (*Delmas et al., 2011*). Piezo1 and Piezo2 are necessary and sufficient for mechanically activated cation channel activity (*Coste et al., 2010*). These proteins are expressed in various mechanically sensitive cell types, and Piezo1 and Piezo2 have recently been shown to be required for vascular development and touch sensing, respectively (*Li et al., 2014*; *Maksimovic et al., 2014*; *Ranade et al., 2014a*; *Ranade et al., 2014b*; *Woo et al., 2014*). In humans, Piezo1 gain-of-function mutations are associated with a hereditary red blood cell condition termed dehydrated hereditary stomatocytosis, while Piezo2 gain-of-function mutations are associated with three phenotypically overlapping conditions termed distal arthrogryposis type 5, Gordon Syndrome, and Marden–Walker Syndrome (Piezo2) (*Albuisson et al., 2013*; *Andolfo et al., 2013*; *Bae et al., 2013*; *Coste et al., 2013*; *McMillin et al., 2014*). Piezo proteins form a distinct class of proteins with no apparent sequence similarity to other proteins and channels (*Coste et al., 2010*; *Bae et al., 2011*; *Nilius and Honore, 2012*). They typically consist of >2000 amino acids with ~30–40 putative transmembrane segments, and Piezo1 has been shown to assemble as a homotetramer of ~1.2 million daltons. Purified Piezo1 can be reconstituted in lipid bilayers resulting in spontaneous cation currents. This indicates that Piezos constitute

**eLife digest** Within our bodies, cells and tissues are constantly being pushed and pulled by their surrounding environment. These mechanical forces are then transformed into electrical or chemical signals by cells. This process is crucial for many biological structures, such as blood vessels, to develop correctly, and is also a key part of our senses of touch and hearing.

In 2010, researchers discovered a group of ion channels—proteins embedded in the membrane that surrounds a cell—that open up when a force is applied and allow ions such as calcium, potassium, and sodium to flow. This movement of ions generates the electrical response of the cell to the applied force. However, not much is known about how these 'Piezo' ion channels work. To investigate this, it is important to be able to precisely control how and when the Piezo channels open. Many other ion channels are studied by using small chemical compounds to activate them, but there were none that were known to act on Piezo proteins.

Syeda et al.—including some of the researchers involved in the 2010 work—screened over three million compounds for their ability to cause calcium ions to flow into human cells to try to identify chemicals that activate the Piezo channels. This revealed one promising candidate named Yoda1, which specifically activated Piezo1: a Piezo protein that had previously been linked to a role in blood vessel development in embryos.

To investigate how Yoda1 activates Piezo1, Syeda et al. placed Piezo1 in an artificial cell membrane that did not contain any other cellular components. When Yoda1 was added to this set up, the Piezo1 channels opened up. This suggests that Piezo1 and Yoda1 interact in a manner that does not require additional cellular components other than a cell membrane.

Separate work by Cahalan, Lukacs et al. uses Yoda1 to reveal that Piezo1 helps to control the volume of red blood cells, showing that in the future, Yoda1 could be valuable in research that investigates the roles of Piezo1.

channel-forming proteins, as opposed to accessory subunits (*Coste et al., 2012*). Both vertebrate and invertebrate Piezo channels can be activated by mechanical stimuli suggesting an evolutionarily conserved gating mechanism geared to transduce mechanical force (*Kim et al., 2012*). Indeed, to date, mechanical stimuli are the only means to activate Piezo ion channels. In comparison, temperature-activated transient receptor potential (TRP) ion channels are polymodal and are the sensors of various chemicals that cause a burning sensation such as capsaicin and mustard oil, as well as endogenous compounds that cause inflammation (*Julius, 2013*). Studies on the chemical activation of TRP channels have been crucial to understand the physiological role of these channels and have contributed to mechanistic appreciation of how these ion channels are gated (*Jordt and Julius, 2002*; *Bandell et al., 2006*; *Macpherson et al., 2007*; *Cao et al., 2013*; *Julius, 2013*). The discovery of a chemical agonist of Piezo channels could thus benefit the study of mechanotransduction.

## Results and discussion

We set out to probe Piezos for chemical-mediated activation. As Piezos are calcium-permeable channels, we hypothesized that Piezo activity could be monitored using calcium-sensitive fluorophores. We tested this by overexpressing Piezo1 in human embryonic kidney (HEK) cells and monitoring intracellular calcium in response to pressure exerted on the cell via a blunt glass probe. In Piezo1 expressing cells, a sequence of mechanical stimulations using a piezo-electric driven probe caused reversible calcium responses that increased with increasing probe displacement until ultimately an irreversible $Ca^{2+}$ signal ensued as a consequence of membrane rupture. In contrast, control cells showed no reversible responses and only irreversible calcium flux upon cell perforation was observed. (*Figure 1A*). This suggested that Piezo1-mediated calcium flux could be measured via calcium fluorophores. With the objective to identify either a Piezo1 or Piezo2 agonist, we co-transfected HEK cells with mPiezo1 and mPiezo2 cDNAs and screened a collection of ~3.25 million low molecular weight (LMW) compounds for their ability to induce calcium influx in these cells. This led us to identify a synthetic compound that elicits $Ca^{2+}$ flux in Piezo1- but not vector-transfected cells; we named this compound Yoda1 (see 'Materials and methods' and *Figure 1D* for screen and Yoda1 details).

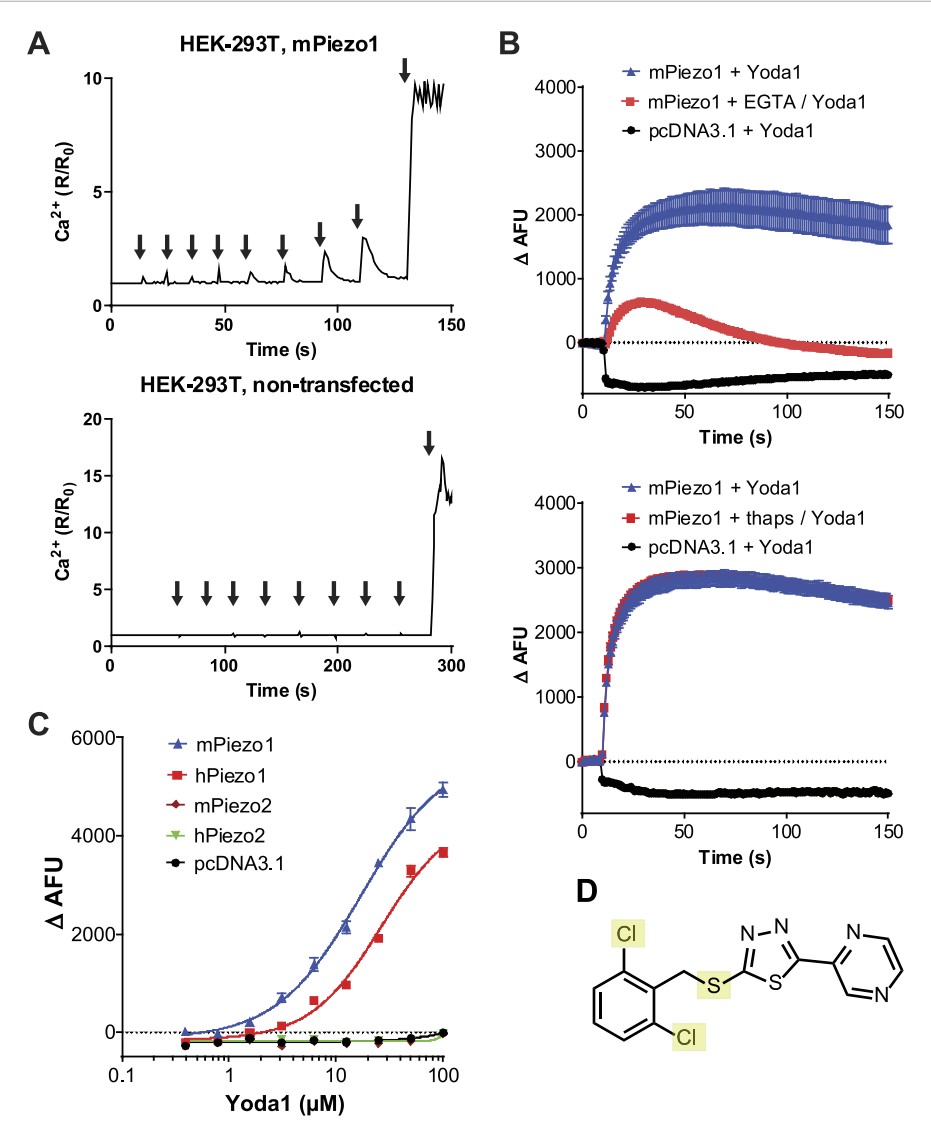

**Figure 1**. A high-throughput screen identifies a Piezo1 activating chemical, Yoda1. (**A**) mPiezo1 mediates Ca²⁺ influx upon mechanical activation. Ratiometric Ca²⁺ imaging (Fura-2) of human embryonic kidney (HEK) 293T cells transiently transfected with Piezo1 or untransfected. Cells were subjected to a series of mechanical stimuli, by pressing a glass probe briefly onto the cell surface for 150 ms (arrows). For each consecutive stimulus, the travel distance of probe was increased by 1 µm (**B**) Yoda1 (25 µM) mediates Ca²⁺ responses (384-well FLIPR) in HEK cells transiently transfected with mPiezo1. When indicated, extracellular calcium was chelated by addition of EGTA, or cells were pretreated with thapsigargin to deplete intracellular calcium stores. Traces represent average ± SEM fluorescence of four wells. (**C**) Concentration-response profiles of mouse and human Piezo1 and Piezo2, transfected HEK293T cells assayed using FLIPR suggesting apparent EC₅₀ of 17.1 and 26.6 µM for mouse and human Piezo1, respectively (95% confidence interval: 13.4 to 21.9, and 20.6 to 34.4), however, compound (in) solubility precludes meaningful conclusions with respect to EC₅₀ (see text). (**D**) Chemical structure of Yoda1. The functional groups tested chlorines and thioether are highlighted.

The following figure supplement is available for figure 1:

**Figure supplement 1**. Piezo1 gain-of-function mutations show increased Yoda1 responses.

Yoda1-induced calcium responses depended largely on calcium influx as the chelation of extracellular calcium dramatically reduced the responses, while the depletion of intracellular calcium stores using thapsigargin did not (*Figure 1B*). Still, calcium-chelating conditions did not completely abolish the responses, raising the possibility of some functional Piezo1 in intracellular membranes upon

overexpression. Concentration-response experiments showed that Yoda1 at micromolar concentrations induced robust $Ca^{2+}$ responses in cells transfected with either human or mouse Piezo1, but not Piezo2-transfected cells, indicating its selectivity for Piezo1 (*Figure 1C*). At higher Yoda1 concentrations (>~20 μM), the solutions became increasingly opaque. Therefore, the apparent $EC_{50}$ is likely affected by compound insolubility and may not allow meaningful interpretation. We further tested six distinct hPiezo1 mutants that we previously identified in xerocytosis patients and which exhibited increased mechanically induced currents (*Albuisson et al., 2013*). Invariably, the Yoda1-induced calcium responses were bigger in cells transfected with these mutant channels than cells transfected with wild-type Piezo1, consistent with their gain-of-function phenotype (*Figure 1—figure supplement 1*). The effect of Yoda1 appears to critically depend on the dichloro substitution (*Figure 1D*), as similar compounds present in the collection lacking the chlorines were not identified in the screen (data not shown). Furthermore, the oxidation state of the thioether group appears critical as no activity could be observed with the sulfone analog (tested at ≤30 μM, data not shown).

We next sought to assess the effect of Yoda1 on mPiezo1 channel function directly by recording mPiezo1-mediated currents, both in the presence and absence of mechanical force (*Figure 2A–E*). In HEK cells transiently transfected with mPiezo1, using a cell-attached patch configuration, currents were measured before and during a series of increasing negative pressures applied via the recording pipette. The presence of Yoda1 caused multiple distinct effects. Firstly, Yoda1 caused a dramatic change in the kinetics of the mechanical responses, as it notably slowed the inactivation phase of the transient currents (*Figure 2A*). Secondly, Yoda1 sensitized mPiezo1 to activation by pressure as indicated by a leftward shift in the current–pressure relationship, reducing the half maximal activation pressure (P50) by about 15 mm Hg (*Figure 2B,C*). Lastly, in the absence of negative pressure, we observed small currents in Yoda1-exposed patches. The Yoda1-dependent currents were a fraction of those attained by stretch: 9.0 ± 2.2% of the maximal attainable current compared to 1.5 ± 0.4% in control patches without Yoda1 (*Figure 2B,D,E*). These results suggest that Yoda1 both modifies Piezo1 mechanotransduction currents and partially activates Piezo1 in the absence of externally applied pressure (note, however, that some membrane tension exists in cell-attached patches even prior to the application of negative pressure) The partial activation of Piezo1 by Yoda1 might be due to a variety of reasons, including (1) an indirect mechanism of action (but see below), (2) an inefficacious gating mechanism (i.e., acting as a gating modifier instead of a full agonist), or (3) a very high actual $EC_{50}$ (see comment on insolubility above). We also tested the effect of Yoda1 (10 μM) in whole-cell configuration, where mechanical pressure can be applied using a piezoelectric-driven glass probe. This concentration did not lead to measurable mPiezo1 currents in the absence of pressure but did cause a clear slowing of the inactivation phase of the mechanically activated currents, similar to the cell-attached patch experiments (*Figure 2F,G*). No such change in kinetics could be observed for Piezo2 consistent with Yoda1 having a Piezo1 selective effect (*Figure 2H*).

As discussed above, Yoda1 might act directly on Piezo1 or indirectly via other membrane or even intracellular mediators. To address this, we set out to test its effect on purified mPiezo1 reconstituted in droplet interface lipid bilayers (DIBs). Previously, we reconstituted purified mPiezo1 channels into asymmetric DIBs containing DiPhytanoyl-sn-glycero-3-PhosphoCholine (DPhPC) and 1,2-dioleoyl-sn-glycero-3-phosphate (DOPA) and observed spontaneous cation-selective channel activity (*Coste et al., 2012*). Here, to study mPiezo1 activation, we utilized symmetric DIBs made up of only DPhPC in which reconstituted mPiezo1 does not show constitutive activity (*Figure 3A*). Application of 1 μM Yoda1-induced discernable single-channel currents with a conductance similar to what we previously observed for spontaneously active mPiezo1 in asymmetric bilayers (*Figure 3B*). Higher Yoda1 concentrations yielded robust currents with a staircase-like appearance, indicating the presence of multiple (30–40) functional channels (*Figure 3C,D*).

Next, we assessed single-channel parameters of mPiezo1 reconstituted in symmetric DPhPC bilayers in the presence of 1 μM Yoda1 (*Figure 3E*). The channel exhibited closed dwell-time distributions that fitted well to a two-component probability density function; $\tau_{1\ closed} = 3 \pm 1$ ms and $\tau_{2\ closed} = 57 \pm 15$ ms. The two time constants differ in duration by > 10-fold, a property of a bursting channel, where $\tau_{1\ closed}$ is the closed time within a burst, and $\tau_{2\ closed}$ is the closed time between bursts. The open dwell time is fitted to a single-component probability density function with characteristic mean open time $\tau_{open} = 55 \pm 9$ ms. Although, no direct comparison can be made for mPiezo1 parameters with and without Yoda1 (due to lack of mPiezo1 activity in symmetric bilayers without Yoda1), we analyzed single-channel

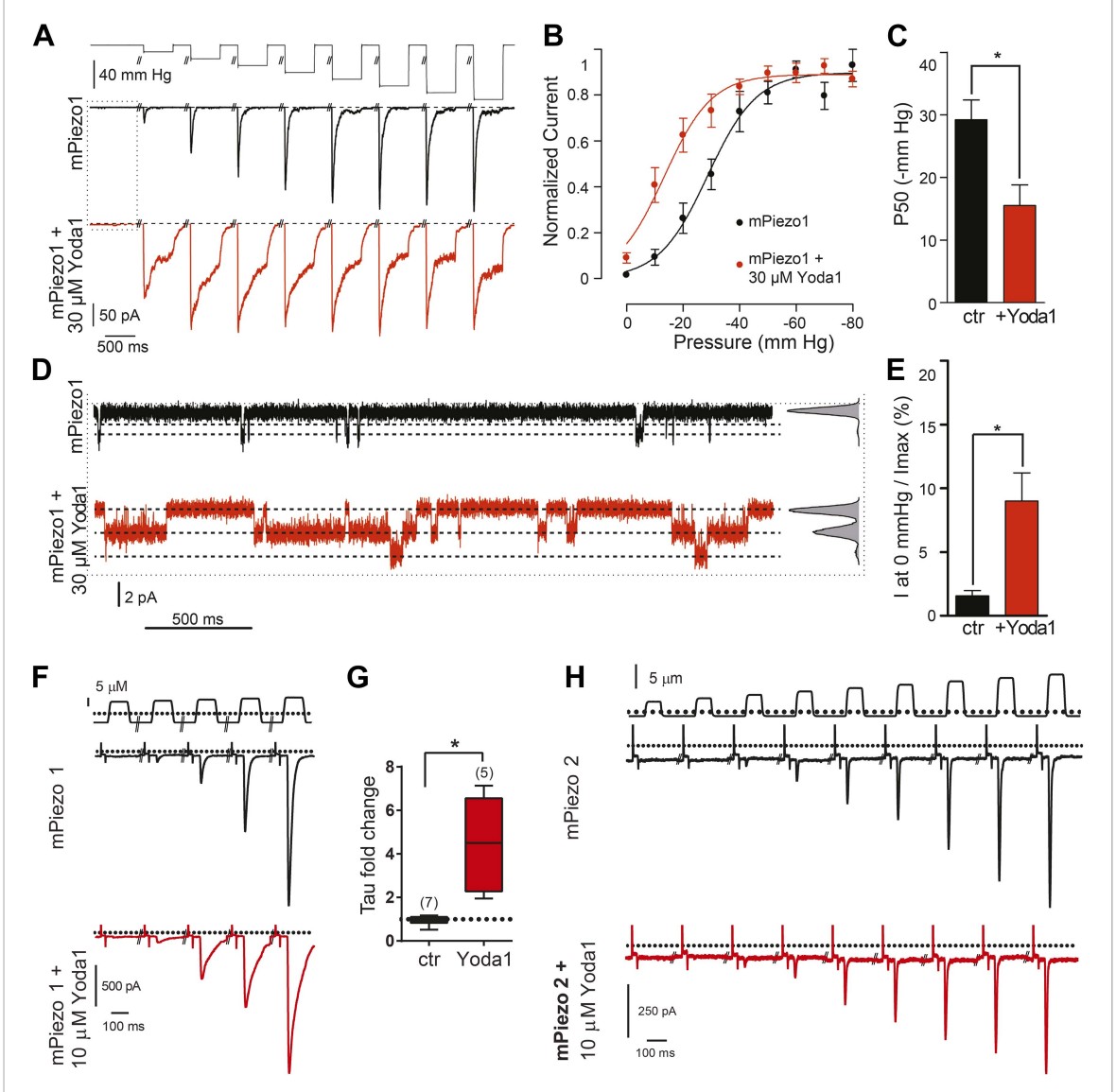

**Figure 2**. Yoda1 functions as a gating modifier of Piezo1. (**A–E**) mPiezo1-transfected HEK293T cells, cell-attached patch configuration. (**A**) Typical recordings of stretch-activated currents at −80 mV in two mPiezo1-transfected cells with or without 30 μM Yoda1 in the patch pipette. Negative pressure pulses from 0 to −80 mm Hg are applied for 500 ms every 15 s. (**B**) Average normalized current–pressure relationships from mPiezo1-transfected cell recordings with or without 30 μM Yoda1 in the patch pipette (n = 8 and 12, respectively). (**C**) Average P50 values from individual cells used for panel B (p < 0.05, Mann–Whitney t-test). (**D**) High magnification of recording traces shown in panel **A** in the absence of stretch stimulation. Left panels are full-trace histograms. (**E**) Average current without stretch stimulation normalized to maximal stretch-activated current from mPiezo1-transfected cells recorded at −80 mV with or without 30 μM Yoda1 in the patch pipette (n = 8 and 12, respectively; p < 0.05, Mann–Whitney t-test). (**F–H**) mPiezo1- and mPiezo2-transfected HEK293T cells, whole-cell configuration. (**F**) Stimulus displacement in 0.5-μm increments every 10 s before (black trace) and 1–2 min after bath application of 10 μM Yoda1 (red trace). A 20-mV step was applied in the beginning of each sweep (sweeps are concatenated and hack marks indicate ∼10 s) to monitor membrane ($R_m$) and access ($R_a$) resistance. (**G**) The fold change in the inactivation time constant indicates a significant slowing of inactivation during Yoda1 exposure. The effect was completely reversible (not shown). The baseline tau prior to Yoda1 exposure was 16.5 ± 1.5 ms (n = 5) (**H**). No effect was observed upon Yoda1 exposure (up to 5 min) to the mechanically activated currents elicited in a cell expressing mPiezo2. Fold change in inactivation time constant was 0.89-, 1.19-, and 1.25-fold (n = 3). Dotted lines indicated 0 current level (current traces) and displacement at which cell was visibly touched (top). *p < 0.005, Mann–Whitney t-test.

properties of mPiezo1 acquired in asymmetric bilayers without Yoda1 (*Figure 3F*). In an asymmetric bilayer, the channel exhibited characteristic bursting pattern with two closed time distributions, $\tau_{1\ closed} = 5 \pm 1$ ms, $\tau_{2\ closed} = 47 \pm 9$ ms, and an open time distribution $\tau_{open} = 13 \pm 4$ ms.

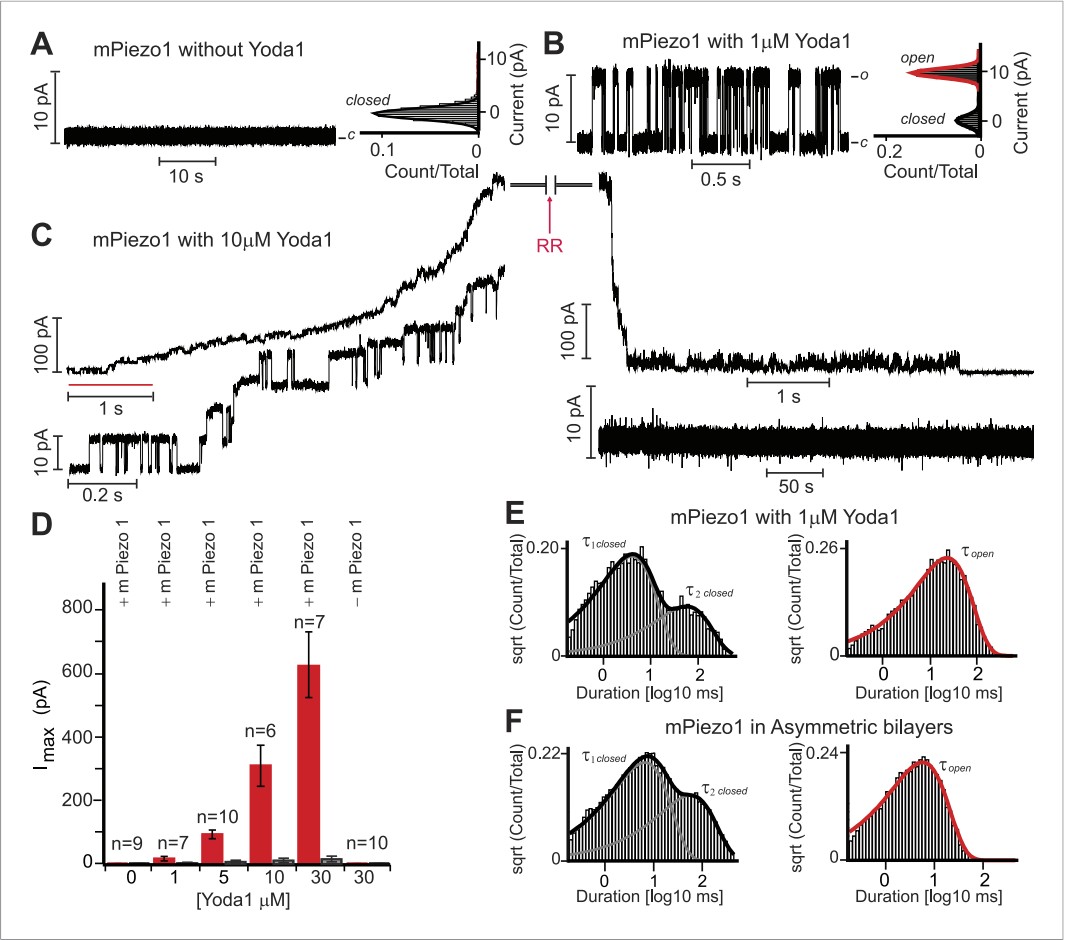

**Figure 3**. Yoda1 activates mPiezo1 in a membrane-delimited fashion. (**A**) Electrical recordings of reconstituted mPiezo1 in the symmetric DiPhytanoyl-sn-glycero-3-PhosphoCholine (DPhC) bilayers and corresponding all point current histograms without the application of Yoda1. (**B**) Single-channel electrical recordings of reconstituted mPiezo1 in the symmetric DPhC bilayers in the presence of 1 μM Yoda1. The calculated single-channel conductance of outward currents from the corresponding all point current histograms is 98 ± 9 pS in 0.5 M KCl, 20 mM HEPES, pH 7.4 at V = 100 mV. (**C**) Macroscopic currents of mPiezo1 in the presence of 10 μM Yoda1 (upper left panel) followed by the injection of 30 μM blocker RR (upper right panel). The lower left panel is an expansion of the record (red line) to highlight multiple-channel openings. The lower right panel shows a complete block of channel activity after 6 s of RR injection. (**D**) Maximum current obtained at the indicated concentrations of Yoda1 (red bars) and the subsequent block by RR (black bars). Each concentration point is plotted (red bars) as the function of maximum currents obtained in an 'n' number of experiments at V = 100 mV. Error bars indicate standard deviation. Note the lack of Piezo activity either without Yoda1 (n = 9) or without mPiezo1 (n = 10) in the bilayers. When indicated, Yoda1 is reconstituted in the DPhC liposomes prior to the bilayer formation. (**E**) Representative histograms of closed (left graph) and open (right graph) dwell times extracted from single-channel analysis of mPiezo1 in the presence of 1 μM Yoda1; $\tau_{1\ closed} = 3 \pm 1$ ms, $\tau_{2\ closed} = 57 \pm 15$ ms, and $\tau_{open} = 55 \pm 9$ ms.
(**F**) Representative histograms of closed (left graph) and open (right graph) dwell times extracted from single-channel analysis of mPiezo1 reconstituted in an asymmetric bilayers (without Yoda1); $\tau_{1\ closed} = 5 \pm 1$ ms, $\tau_{2\ closed} = 47 \pm 9$ ms, and $\tau_{open} = 13 \pm 4$ ms.

The significant difference in $\tau_{open}$ (13 ± 4 ms vs 55 ± 9 ms) suggests that mPiezo1 remains open for longer times in the presence of Yoda1. No significant change was observed in $\tau_{closed}$, suggesting that Yoda1 mainly stabilizes the open state rather than destabilizing the closed state.

A detailed biophysical analysis will be required to understand the mechanism of action of Yoda1. Our present analysis provides the first steps towards this goal. For instance, our lipid bilayer experiments suggest that Yoda1 does not require other proteins or specific lipid domains to exert an

effect on Piezo1. This suggests its effect is either directly on the channel or via long-range membrane-delimited effects, for instance through a change in membrane tension or curvature of the membrane. Indeed, compounds that modify membrane curvature are known to affect mechanically sensitive ion channels (*Patel et al., 1998*). However, such compounds are typically amphipaths which Yoda1 is not (*Sheetz and Singer, 1974*). More importantly, the effect of Yoda1 appears governed by stringent structural requirements both on the side of the chemical and on the side of the channel (as no effect on Piezo2 was observed), fitting a model in which Yoda1 directly interacts with Piezo1.

Our electrophysiological experiments in cells suggest that Yoda1 prominently affects the sensitivity and the inactivation kinetics of mechanically induced responses but at best causes a slight mPiezo1 activation in the absence of mechanical stimuli. In the bilayer system, we observed that Yoda1 stabilizes the open channel, potentially explaining the slowing of mPiezo1 inactivation kinetics observed in cells. However, we also observe prominent Yoda1-dependent calcium responses in cell culture and currents in artificial bilayers in the absence of externally applied forces. Therefore, the discrepancy in various levels of channel activity observed with different assays used here remains unexplained, and future in-depth understanding of mechanism of Yoda1 action on Piezo1 might shed light on these apparently disparate observations. Regardless, we show in an accompanying paper that Yoda1 causes Piezo1-dependent red blood cell dehydration, arguing for sufficient activation of the ion channel in the absence of external forces to cause a robust physiological impact.

Irrespective of whether Yoda1 acts as a full activator or as a positive modulator, our results suggest that we have identified the first Piezo1 agonist. This finding is important from two perspectives. Firstly, our studies provide the first evidence of non-mechanical activation of a Piezo channel, suggesting that Piezo1 gating does not exclusively depend on changes in mechanical force. This is important, since it raises the possibility that endogenous agonists of Piezo1 exist which may play an important role in modulating mechanotransduction. Secondly, Yoda1 will provide a valuable tool to facilitate studies aimed at elucidating Piezo1 gating mechanism and exploring its functional significance in various biological processes (see for instance accompanying paper).

## Materials and methods

### Ratiometric calcium imaging

#### Cell culture and transient transfection
HEK293T cells were grown in Dulbecco's modified Eagle's medium containing 4.5 mg/ml glucose, 10% fetal bovine serum, 1× antibiotics/antimycotics. Cells were plated onto 12-mm round glass poly-D-lysine coated coverslips placed in 24-well plates and transfected using Fugene6 (Roche, Basel, Switzerland) according to the manufacturer's instruction. mPiezo1-IRES GFP was transfected at 250 ng/ml. Cells were tested 2 days post transfection.

Ratiometric imaging was performed essentially as described (*Ranade et al., 2014a*). In brief, cells were washed with assay buffer (1× HBSS, 10 mM HEPES, pH7.4) and incubated with 2.5 µM Fura-2 and 0.05% Pluronic F-127 (Life Technologies, Grand Island, NY), in assay buffer for ~30 min after which cells were washed again with assay buffer and fluorescence was monitored at excitation wavelengths alternating between 340 and 380 nm, using an inverted fluorescence microscope/camera/light source combination. Mechanical pressure was exerted on the cells using a fire-polished glass pipette essentially as described (*Coste et al., 2010*). The pipette was moved towards the cell at a speed of 1 mm/s.

### Fluorescent imaging plate reader

#### Cell culture and transient transfection
HEK293T cells were grown in Dulbecco's modified Eagle's medium containing 4.5 mg/ml glucose, 10% fetal bovine serum, 1× antibiotics/antimycotics or penicillin/streptomycin. Cells were seeded in poly-D-lysine coated 384-well plates ($1.2 \times 10^4$ cells/well) or 1536-well plates ($1 \times 10^4$ cells/well) and simultaneously transfected using Fugene6 (Promega, Madison, WI) per manufacturer's instructions and 62.5 ng cDNA in 40 µl media/well (384 well) or 6.8 ng cDNA in 4 µl media/well (1536 well).

#### 384-well format
2 days after transfection, the cells were washed with assay buffer (1× HBSS, 10 mM HEPES, pH7.4) using a ELx405 CW plate washer (BioTek, Winooski, VT). Cells were incubated with assay buffer containing 4 µM Fluo3 and 0.04% Pluronic F-127 (Life Technologies) for ~60 min and then washed

again with assay buffer. Fluorescence was monitored on a fluorescent imaging plate reader (FLIPR) Tetra. To chelate extracellular calcium (1× HBSS contains 1.26 mM $CaCl_2$), 2 mM ethylene glycol tetraacetic acid (EGTA) was added to the cells 1 min before addition of the indicated Yoda1 concentration in presence of 2 mM EGTA. To deplete intracellular calcium, 7.5 µM thapsigargin was added 15 min before Yoda1 addition. A 10-mM stock solution of Yoda1 in dimethyl sulfoxide (DMSO) was used resulting in a maximum of 1% DMSO in the assay. Concentration-response curves were fitted using a sigmoidal dose–response at variable slope (GraphPad Prism, La Jolla, CA).

## 1536-well high-throughput screen format

With objective of identifying either a Piezo1 or Piezo2 agonist, we co-transfected cells with mPiezo1 and mPiezo2 cDNA at equal amounts. 2 days after, transfection cells were incubated with Calcium5 (Molecular Devices, Sunnyvale, CA) according to manufacturer's instruction and fluorescence was monitored on FLIPR Tetra. About 3.25 million compounds from the LMW Novartis screening library, which includes public domain and proprietary drug-like molecules, were screened at a concentration of 5 µM. Approximately 9000 hits, as defined by 50% activation above DMSO control wells, were selected for retesting in co-transfected cells as well as individual Piezo1 and 2 transfection and control cells. From this, Yoda1 was identified as a potential Piezo1 activator and selected for further study. Yoda1 was obtained from Maybridge Chemical Company.

## Cell-attached patch clamp recordings

### Cell culture and transient transfection

HEK293T cells were grown in Dulbecco's modified Eagle's medium containing 4.5 mg/ml glucose, 10% fetal bovine serum, 50 units/ml penicillin, and 50 µg/ml streptomycin. Cells were plated onto 12-mm round glass poly-D-lysine coated coverslips placed in 24-well plates and transfected using Lipofectamine 2000 (Invitrogen, Carlsbad, CA) according to the manufacturer's instruction. mPiezo1-IRES GFP plasmid was transfected at a concentration of 600 ng/ml. Cells were recorded from 12 to 48 hr post transfection.

### Cell-attached patch clamp recordings

Stretch-activated currents were recorded using Axopatch 200B amplifier (Molecular Devices Axopatch 200B). Currents were sampled at 20 kHz and filtered at 2 kHz. External solution used to zero the membrane potential consisted of (in mM) 140 KCl, 1 $MgCl_2$, 10 glucose, and 10 HEPES (pH 7.3 with KOH). Recording pipettes were of 2–3 MΩ resistance when filled with standard solution composed of (in mM) 130 NaCl, 5 KCl, 1 $CaCl_2$, 1 $MgCl_2$, 10 TEA-Cl, and 10 HEPES (pH 7.3 with NaOH). When specified, pipette solution was supplemented with 30 µM Yoda1. Membrane patches were stimulated with 500-ms negative pressure pulses through the recording electrode using Clampex controlled pressure clamp HSPC-1 device (ALA-Scientific, Farmingdale NY). Consecutive sweeps with pressure stimulation ranging from 0 to −80 mm Hg (Δ-10 mm Hg) were applied every 15 s. Full-trace histograms in *Figure 2D* were fitted with Gaussian equations using multi-peak fitting analysis of IGOR Pro software.

## Whole-cell patch clamp recordings

Mechanically activated whole-cell currents at a holding potential of −80 mV were elicited by indentation by a blunt glass probe as described (*Coste et al., 2010*). Application of vehicle or compounds was achieved by puffer pipette as described (*Dubin et al., 1999*) or bath application; results were similar and combined. Voltage ramp-induced currents were recorded as described (*Dubin et al., 1999*).

### Analysis

The fold change in inactivation time constant (fitted with stimulus vs response with variable slope; GraphPad Prism v6) in the presence of vehicle or compound was determined as described (*Dubin et al., 2012*).

## Droplet lipid bilayer recordings

### Single-channel recordings using droplet lipid bilayers

Liposomes were made from DPhPC as previously described (*Bayley et al., 2008*; *Syeda et al., 2008*). For reconstitution, the purified protein (protein purified as described [*Coste et al., 2012*]) was diluted ~200–400-fold into preformed liposomes suspension in 0.5 M KCl, 20 mM HEPES pH 7.4.

For Yoda1 containing liposomes, Yoda1 stock of 10 mM was made in DMSO and further diluted to give 1–30 μM final concentration in DPhPC liposomes. The liposomes were extruded through 0.1-mm filter (Avanti Polar Lipids, Alabaster, AL). The protein was added in the Yoda1-containing liposomes prior to the single-channel recordings. All the experiments were performed in 100% DPhPC symmetric bilayers (with and without Yoda1), in 0.5 M KCl to potentiate signal-to-noise ratio. Electrode carrying the proteoliposome droplet was connected to the working end of the amplifier head stage (Molecular Devices Axopatch 200B). The second electrode, in a droplet containing the DPhPC liposomes, was connected to the grounded end of the head stage. In the indicated experiments, RR was injected to a final concentration of 30 μM using a Nano injector (VWR instrument, Sugar Land, TX).

## Single-channel acquisition and analysis

Single-channel acquisition and analysis were performed as described previously (*Coste et al., 2012*). Segments of continuous recordings in the range of $50\ s \leq t \leq 500\ s$ were used for analysis. The currents were sampled at 20 kHz and filtered at 2 kHz. Additional offline filtering of 1 kHz was applied to the recordings for display. Conductance was determined by fitting a Gaussian curve to the single channel all point current amplitude histograms. Event detection was performed by time-course fitting with the segmental K means (SKM) implemented in QuB software. To avoid the detection of erroneous events, the receiver dead time ($t_d$) was set at 300 μs for all records. Therefore, transitions shorter than $t_d$ were ignored; transitions longer than $t_d$ were accepted as events. Open dwell-time distributions are fitted with a single-component probability density function, whereas closed dwell-time distributions are fitted with a two-component probability density function implemented in QuB. The calculated values are reported as mean ± standard deviation, n denote number of experiments.

## Acknowledgements

We thank Bailong Xiao for mPiezo1 protein isolation. This work was supported by NIH NS083174 to AP, and NIH GM49711 to MM. AP is a Howard Hughes Medical Institute Investigator.

## Additional information

### Funding

| Funder | Grant reference | Author |
| --- | --- | --- |
| Howard Hughes Medical Institute (HHMI) | | Ardem Patapoutian |
| National Institutes of Health (NIH) | NIH NS083174 | Ardem Patapoutian |
| National Institutes of Health (NIH) | NIH GM49711 | Mauricio Montal |

The funders had no role in study design, data collection and interpretation, or the decision to submit the work for publication.

### Author contributions

RS, AED, BC, MB, Conception and design, Acquisition of data, Analysis and interpretation of data, Drafting or revising the article; JX, Conception and design, Acquisition of data, Analysis and interpretation of data; JM, TH, JM, JL, Acquisition of data, Analysis and interpretation of data; DCT, AP, Conception and design, Analysis and interpretation of data, Drafting or revising the article; IHE, HMP, AMS, MM, Conception and design, Analysis and interpretation of data

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
