## [Decision Letter]

Thank you for sending your work entitled “Chemical activation of the mechanotransduction channel Piezo1” for consideration at *eLife*. Your article has been favorably evaluated by Janet Rossant (Senior editor), a Reviewing editor, and three reviewers. Two of the reviewers, Eric Honoré and Jon Levine, have agreed to share their identity.

The Reviewing editor and the reviewers discussed their comments before we reached this decision, and the Reviewing editor has assembled the following comments to help you prepare a revised submission. We are including the three reviews (lightly edited) at the end of this letter, as there are some specific and useful suggestions in them that will not be repeated in the summary here. All of the reviewers were impressed with the importance and novelty of your work.

Reviewer #1:

In the manuscript “Chemical activation of the mechanotransduction channel Piezo1”, the authors report the first chemical agonist (Obi1) of the mechanically activated ion channel Piezo1. The molecule was identified by an unbiased screen of an impressive ∼3 million compound library. The authors use a combination of calcium-imaging, detailed single channel and whole-cell and artificial bilayer electrophysiology to convincingly show that Obi1 is in fact an agonist acting specifically on Piezo1, but not Piezo2.

This finding is important, because it is the first demonstration that this mechanically activated ion channel can also be gated chemically. Due to the astronomical number of screened molecules and the fact that only one hit was identified this molecule is highly unlikely to be identified otherwise and it highlights how unique it is (and probably will be). Having this molecule is significant to study Piezo-related mechanisms and the authors demonstrate this nicely in the accompanying manuscript.

I find this study very impressive and highly interesting. The authors go beyond any reasonable effort in order to convincingly show how Obi1 acts on Piezo1 mechanistically. Thus I have no suggestions for improvements and only a few minor points.

1) Figure 3: It is better to plot the square root of counts instead of counts as this leads to normalization of error bars (although not plotted) in all bins.

2) The chemical structure of Obi1 should be in the main manuscript and not the supplement and the two functional groups tested (chloridine and thioether) should be highlighted.

Reviewer #2:

This interesting manuscript reports the discovery of a specific opener for the mechano-sensitive ion channel Piezo1. The Obi1 compound shifts the pressure-effect curve leftward and removes inactivation. Remarkably, this opener works after reconstitution of Piezo1 into an artificial bilayer. Those findings are conceptually important as it demonstrates that this mammalian stretch-activated ion channel can also be opened by an agonist in the absence of mechanical stress. This study also suggests that a natural opener of Piezo1 may possibly exist. I believe that this report is important for a better understanding of cellular mechanotransduction and will certainly be a highly cited paper.

The discovery of the Piezo channels, of their function and pharmacology represent a major step in our understanding of molecular mechanotransduction. I only have a few suggestions which may help to further improve this manuscript. However, those points are optional and I feel that the present paper could be published in its present form.

Minor points and suggestions:

1) Why does fluorescence goes down in the control cells (pcDNA3.1 + Obi1), as shown in Figure 1 and Figure 1?

2) There is no basal Piezo1 opening in a reconstituted symmetric bilayer, unlike with asymmetric bilayer as previously reported by the same authors. Can you apply mechanical stress on the symmetric bilayer and now demonstrate that force gates the purified and reconstituted Piezo1?

3) Is the effect of Obi1 reversed by the peptide GsTMx-4, which is also a gating modifier?

4) Is the effect of Obi1 mimicked by the anionic amphipathic crenator trinitrophenol? (Although, Obi1 does not work on Piezo2 and a non-specific bilayer effect seems unlikely. Nevertheless, one may imagine that Piezo1 might be more sensitive than Piezo2 to bilayer stress).

5) In the same line, is the effect of Obi1 reversed by the amphipathic cup-former chlorpromazine?

6) Of course a key point would be the discovery of natural opener for Piezo1. Any idea?

7) Please indicate the charge of the molecule at a physiological pH (Figure 1—figure supplement 1).

8) Can you please cite the review of Nilius-TINS about Piezo1?

9) Please cite the work of Sachs and collaborators, reporting the modulation of Piezo1 by the gating modifier GsTMx-4?

Reviewer #3:

The Patapoutian group, that discovered Piezo 1 and 2, have begun the long hard process of drugging these ion channels. In this first manuscript to come from this work, they have successfully isolated a Piezo 1 (and not Piezo 2) agonist, from a library of 2.5 million compounds. Given the rapidity with which the new field of Piezo biology has developed, this compound will be immensely useful to a growing number of investigators interested in the role of Piezo 1 in physiology and pathophysiology.

---

## [Author Response]

We thank the reviewers for their constructive input. We address their remarks and concerns below. Initially we named our compound Obi1. However, while our manuscript was in review, a similarly named compound came to our attention (Piezo is not the target of this other OB1). However, to avoid confusion, we have renamed our compound Yoda1.

Reviewer #1:

*1)*
Figure 3*: It is better to plot the square root of counts instead of counts as this leads to normalization of error bars (although not plotted) in all bins*.

Figure 3 and Figure 3 has now been plotted as a square root of counts.

*2) The chemical structure of Obi1 should be in the main manuscript and not the supplement and the two functional groups tested (chloridine and thioether) should be highlighted*.

The chemical structure of the compound Yoda1 is now in the main figure as Figure 1 and the two functional groups are highlighted.

Reviewer #2:

*1) Why does fluorescence goes down in the control cells (pcDNA3.1 + Obi1), as shown in*
Figure 1
*and*
Figure 1*?*

We observe this effect not only in control cells but also in the Piezo transfected cells when vehicle without Yoda1 is pipetted on the cells. We therefore conclude this as a non-specific effect.

2) There is no basal Piezo1 opening in a reconstituted symmetric bilayer, unlike with asymmetric bilayer as previously reported by the same authors. Can you apply mechanical stress on the symmetric bilayer and now demonstrate that force gates the purified and reconstituted Piezo1?

We are working on the mechanical activation of reconstituted Piezo1. We believe this to be beyond the scope of the current work.

3) Is the effect of Obi1 reversed by the peptide GsTMx-4, which is also a gating modifier?

In past experiments we did not observe robust block of Piezo1 mechanoresponses with commercially available GsMTx4. We have to request GsMTx4 from the F. Sachs lab to revisit this at a later time.

*4) Is the effect of Obi1 mimicked by the anionic amphipathic crenator trinitrophenol? (Although, Obi1 does not work on Piezo2 and a non-specific bilayer effect seems unlikely. Nevertheless, one may imagine that Piezo1 might be more sensitive than Piezo2 to bilayer stress)*.

Trinitrophenol (TNP) causes activation of TREK-1 at concentrations of 100-400µM (Patel et al. 1998). Application of 100 or 400µM TNP did not induce Piezo1 specific calcium responses (measured by FLIPR) nor did it block Piezo1 mediated Ca^2+^ responses to 40µM Yoda1 (data not shown).

*5) In the same line*, *is the effect of Obi1 reversed by the amphipathic cup-former chlorpromazine?*

Chlorpromazine (CPZ) at concentrations of 0.1-10 µM was shown to block TREK-1 channels (Patel et al. 1998). Application of 0.1 or 10µM CPZ did not induce Piezo1 specific calcium responses (measured by FLIPR) nor did it block Piezo1 mediated Ca^2+^ responses to 40µM Yoda1 (data not shown).

*6) Of course a key point would be the discovery of natural opener for Piezo1*. *Any idea?*

We hope future investigations may yield insight into this interesting question.

*7) Please indicate the charge of the molecule at a physiological pH (*Figure 1—figure supplement 1*)*.

The compound is not charged at neutral pH (Figure 1).

8) Can you please cite the review of Nilius-TINS about Piezo1?

This citation was added.

9) Please cite the work of Sachs and collaborators, reporting the modulation of Piezo1 by the gating modifier GsTMx-4?

This citation was added.